

# Changes in nutrients and decay rate of *Ginkgo biloba* leaf litter exposed to elevated O₃ concentration in urban area

Wei Fu[1,2], Xingyuan He[1], Sheng Xu[1], Wei Chen[1], Yan Li[1], Bo Li[1], Lili Su[1] and Qin Ping[1]

[1] Key Laboratory of Forest Ecology and Management, Institute of Applied Ecology, CAS, Shenyang, China
[2] University of Chinese Academy of Sciences, Beijing, China

Corresponding author
Sheng Xu, xusheng@iae.ac.cn,
shengxu703@126.com

## ABSTRACT

Ground-level ozone (O₃) pollution has been widely concerned in the world, particularly in the cities of Asia, including China. Elevated O₃ concentrations have potentially influenced growth and nutrient cycling of trees in urban forest. The decomposition characteristics of urban tree litters under O₃ exposure are still poorly known. *Ginkgo biloba* is commonly planted in the cities of northern China and is one of the main tree species in the urban forest of Shenyang, where concentrations of ground-level O₃ are very high in summer. Here, we hypothesized that O₃ exposure at high concentrations would alter the decomposition rate of urban tree litter. In open-top chambers (OTCs), 5-year-old *G. biloba* saplings were planted to investigate the impact of elevated O₃ concentration (120 ppb) on changes in nutrient contents and decomposition rate of leaf litters. The results showed that elevated O₃ concentration significantly increased K content (6.31 ± 0.29 vs 17.93 ± 0.40, $P < 0.01$) in leaves of *G. biloba*, significantly decreased the contents of total phenols (2.82 ± 0.93 vs 1.60 ± 0.44, $P < 0.05$) and soluble sugars (86.51 ± 19.57 vs 53.76 ± 2.40, $P < 0.05$), but did not significantly alter the contents of C, N, P, lignin and condensed tannins, compared with that in ambient air. Furthermore, percent mass remaining in litterbags after 150 days under ambient air and elevated O₃ concentration was 56.0% and 52.8%, respectively. No significant difference between treatments was observed in mass remaining at any sampling date during decomposition. The losses of the nutrients in leaf litters of *G. biloba* showed significant seasonal differences regardless of O₃ treatment. However, we found that elevated O₃ concentration slowed down the leaf litter decomposition only at the early decomposition stage, but slightly accelerated the litter decomposition at the late stage (after 120 days). This study provides our understanding of the ecological processes regulating biogeochemical cycles from deciduous tree species in high-O₃ urban area.

## INTRODUCTION

In recent decades, due to the large increases in the emission of O₃ precursors including NO$_x$ and VOCs around the world, the ground-level O₃ concentrations are constantly increasing, particularly in Asia (*Sitch et al., 2007*; *IPCC, 2013*). It is estimated that the O₃ concentration in the troposphere will increase by 40% to 60% by 2100 (*Akimotoa et al.,*

*2015*). Among air pollutants, $O_3$ has the most widespread negative impact on terrestrial vegetation. In particular, it may potentially influences on biogeochemical cycles of forest ecosystems (*Nikolova et al., 2010*; *Calatayud et al., 2011*; *Sicard et al., 2016*).

Effects of $O_3$ on forest ecosystems productivity and feedbacks have been widely investigated worldwide (*Chappelka & Samuelson, 1998*; *Paoletti, 2006*; *De Bauer & Hernández-Tejeda, 2007*) and recently were reviewed (*Wang et al., 2016*). In contrast, $O_3$ effects on litter decomposition are much less known (*Nikula, Vapaavuori & Manninen, 2010*). Indirect evidence comes from litter photodegradation studies in semi-arid and arid ecosystems (*Austin & Vivanco, 2006*). Among the few studies carried out in natural forest ecosystems, *Parsons, Bockheim & Lindroth (2008)* showed that, over a 23-months observation period on leaf litterbags of aspen and birch reciprocally transplanted to separate the effect of substrate quality from environment effects, increasing $O_3$ concentration by fumigation slowed down both aspen and birch litter decay rate, exacerbating the effects of elevated $CO_2$ concentration, but accelerated birch litter decay under ambient $CO_2$. A negative effect of $O_3$ fumigation on litter decay rate was also observed for holm oak leaf litter in Mediterranean forest (*Baldantoni, Fagnano & Alfani, 2011*). Such observations were explained by $CO_2$- and $O_3$-mediated changes in litter chemistry, particularly carbohydrates, nitrogen, and tannins. $O_3$ effects on litter decomposition in urban forests have not yet been explored. Filling such gap is particularly important, as in urban ecosystems, where tropospheric $O_3$ concentration can be very high due to photochemical air pollution; urban trees play a fundamental role in mitigating air pollution (*Manes et al., 2012*). Their leaf litter, if decaying faster when exposed to high $O_3$ concentration, would improve soil chemical properties and promote nutrient cycles, therefore affecting the sustainable development of urban areas (*Nikula, Vapaavuori & Manninen, 2010*; *Xu et al., 2012*).

*Ginkgo biloba* is commonly planted in the cities of northern China and it is one of the main tree species in the urban forest of Shenyang, Liaoning Province, China. According to our recent observations, the highest $O_3$ concentration at ground level is frequently over 40 ppb or even up to more than 80 ppb during the summer in the urban area of Shenyang city (*Xu et al., 2015*). For many years, we assessed the effects of elevated $O_3$ concentration on the eco-physiology of urban trees including *G. biloba* (*He et al., 2009*; *Lu et al., 2009*; *Li et al., 2011*; *Xu et al., 2015*). Here we aim to complement such previous studies with a manipulative experiment testing $O_3$ effects on *G. biloba* leaf litter decomposition and chemical features. Based on the results of many previous studies, we hypothesized that $O_3$ exposure at high concentrations commonly experienced by urban trees, could alter chemical composition and decrease the decomposition rate of this tree leaf litter. In this study, we predicted that elevated $O_3$ concentration would change the chemical compositions of leaves and decomposition rate of leaf litter. Therefore, the main objectives of this study are (1) to assess the changes of leaf litter quality of *G. biloba* fumigated by elevated $O_3$ concentration, including the changes in the contents of some nutrients and secondary metabolites, and (2) to evaluate the decomposition rates of leaf litter from this gymnosperm tree species exposed to high $O_3$ concentration.

## MATERIALS & METHODS

### Study site and experimental treatments

The study was conducted in the Shenyang Arboretum of the Chinese Academy of Sciences (41°46′N, 123°26′E) located in an urban area (*He et al., 2009*). The arboretum with a mean elevation of 41 m and an area of 5 ha was founded in 1955, mainly planted with native tree species. There are more than 300 tree species and the forest coverage rate was 53.7% (*He et al., 2016*). Nowadays, it is a near-natural urban forest (*He et al., 2003*). Affected by warm temperate-zone semi-humid monsoon climate, this area has an annual average temperature of 6.2 to 9.7 °C. The average temperatures in January and July are −12.6 and 27.5 °C, respectively. The maximum temperature is 38.3 °C and the minimum temperature is −30.5 °C (*Xu et al., 2014*). The average annual precipitation is 755.4 mm. The frost-free period lasts for 150 d yearly (*Xu et al., 2005*). The flora of the arboretum located area belongs to the intersection of Changbai Mountain, Northern China, and the Mongolian Floras (*Xu et al., 2006*).

This experiment was carried out in open-top chambers (OTCs). Chambers are 4 m in diameter and 3 m in height, with a 45° sloping frustum and 4-m distance between neighbouring OTCs (*Li et al., 2011*; *Xu et al., 2015*). This experiment included control in ambient air (AA, about 40 ppb) and elevated $O_3$ concentration (EO, 120 ppb). AA and EO had three independent OTC replicates with three OTCs, respectively. Six OTCs were used in total. In 20 May, 2012, nine healthy and uniform five-year-old saplings of *G. biloba* (1.5 m in average height) from a local nursery were selected and planted in each OTC. During the growing season, the saplings were irrigated twice a week and fertilized once at the beginning of the experiment. After 60 d (20 July), the saplings were fumigated with $O_3$ for 8 h a day from 9:00-17:00, except in bad weather such as thunderstorm conditions. By the end of fumigation on 5 November, 2012, all the yellow (senescent) leaves of *G. biloba* from each chamber were harvested and regarded as leaf litter. These senescent leaves under AA and EO were randomly divided into two parts: one part was dried at 65 °C to constant weight for determining the initial chemical compositions and the other was sufficiently dried, and stored in a big glass container and then kept at a cool place under room temperature for the litter decomposition experiments in OTCs next year (2013).

The leaf litters were decomposed according to the litterbag method (*Zhang et al., 2015*) by placing them in nylon bags (20 cm × 25 cm) with 1 $mm^2$ mesh. Before the experiment, each bag was filled with 8 g samples and numbered. On 19 May, 2013, 15 bags were placed into each OTC to touch the topsoil and separate them at 10-cm intervals each other. A total of 45 bags of AA and EO were prepared, respectively. Before putting the litter bags into OTCs, the topsoil (0–10 cm) of each treatment was gathered after removing weeds and other sundries from the ground. Soil samples were mixed uniformly after the stones, roots and other debris were removed. Homogenized fresh soil was passed through a sieve (pore size was 5 mm) and dried under natural conditions. The basic physical and chemical properties of topsoil (0–10 cm) are shown in Table 1. $O_3$ fumigation was carried out for 8 h each day (9:00–17:00). The samples were collected once a month (Sampling date is 19 every month from June to October) to be decomposed for 150 days in total. $O_3$
**Table 1** The physical and chemical properties of topsoil (0–10 cm) in OTCs before leaf litter decomposition of *G. biloba*.

| Treatments | pH | SWC (%) | ST (°C) | SOC (mg/g) | C (mg/g) | N (mg/g) | P (mg/g) | K (mg/g) | C/N | C /P |
|---|---|---|---|---|---|---|---|---|---|---|
| AA | 6.75 (0.06) | 28.25 (1.08) | 25.50 (0.82) | 9.12 (0.30) | 33.23 (0.79) | 2.54 (0.03) | 4.44 (0.09) | 5.70 (0.08) | 13.10 (0.40) | 7.48 (0.27) |
| EO | 6.67 (0.03) | 28.34 (0.97) | 25.30 (1.15) | 9.20 (0.04) | 34.12 (0.19) | 2.58 (0.01) | 4.35 (0.06) | 5.44 (0.43) | 13.22 (0.11) | 7.85 (0.06) |

Notes.

Data are shown mean and standard deviation (SD) in the parenthesis ($n = 3$).

AA, ambient air (control); EO, elevated O3; SWC, soil water content; ST, soil temperature; SOC, soil organic matter content.

concentrations were monitored every day throughout the growing period (19 May–20 October 2013), at 0.80 m above the ground, in the OTCs of both AA and EO (Fig. 1). The microclimatic conditions in the OTCs during litter decomposition are summarised in Table 2. After washing and drying, the cleaned litters were placed into Kraft envelope bags and oven-dried at a constant temperature of 65° C. Afterwards, leaf litters were weighed and dry mass were pulverised by a high-speed grinding machine (FW-100; Taisite Instrument Co., Ltd., Tianjing, China) and filtered through a sieve (80 meshes, mesh size is 5 mm). The powdered samples were preserved in self-sealing bags for the laboratory measurements.

## Chemical analyses and experimental statistics

Carbon (C) and nitrogen (N) contents were determined by an elemental analyser (Vario MACRO Cube, Elementar, Germany). Phosphorus (P) and potassium (K) contents were measured by atomic absorption spectrometry (AA800; Perkin Elmer, San Jose, CA, USA) according to the Mo-Sb colorimetric method and flame photometric method, respectively. Lignin content was determined using the ultraviolet spectrophotometric method (*Liyama & Wallis, 1988*) and total phenol content was measured according to the method of *Julkunen-Tiitto (1985)* with a minor modification. The condensed tannins and soluble sugar contents were determined by spectrophotometer (UV-1800, Shimadzu Corp., Kyoto, Japan) according to the butyl alcohol-hydrochloric acid method (*Porter, Hrstich & Chan, 1986*) and anthrone colorimetric method (*Dubois et al., 1956*), respectively.

Mass remaining of leaf litter over time, expressed as percentage of the initial value, was calculated according to the assumed simple exponential model formulated by *Olson (1963)*.

$$Y = A_t/A_0 = e^{-kt}.$$

Where, $Y$, $A_0$ and $A_t$ indicate the remaining rate of litter mass monthly, the initial litter mass (g), and the remaining mass (g) of leaf litter at time $t$ (months), respectively. In addition, e, and $k$ are the base of natural logarithms, and the decomposition coefficient of the litters, respectively. As for $t$, it is the decomposition time (months) including the half-life of decomposition ($t_{0.5} = ln(2)/k$).

The remaining rate of nutrient composition was calculated (*Pancotto et al., 2003*):

$$E = [(M_t \times C_t)/(M_0 \times C_0)] \times 100\%$$

Where, $E$, $M_t$, $M_0$, $C_0$, and $C_t$ represent the remaining (% initial) of nutrient elements, dry mass (g) of leaf litter at the designated time of decomposition, initial dry mass (g), initial
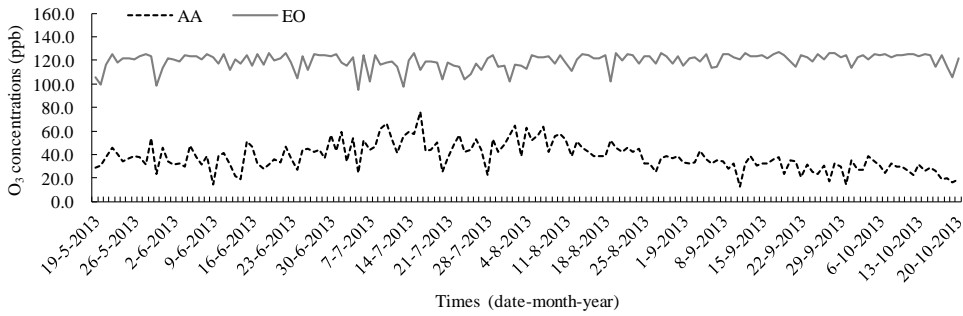

**Figure 1** Seasonal variations in $O_3$ concentrations in OTCs with ambient air (AA) and elevated $O_3$ (EO) during leaf litter decomposition.

**Table 2** Microclimatic conditions in OTCs during gas fumigation in 2013.

| Treatments | $[O_3]_{mean}$ | $[O_3]_{max}$ | AOT40$^{(150)}$ | $RH_{mean}$ | $T_{mean}$ | $[CO_2]$ | DPPFD |
|---|---|---|---|---|---|---|---|
| AA | 38.2 | 76.5 | 1,167.5 | 68.3 | 23.8 | 372.4 | 46.9 |
| EO | 119.5 | 126.0 | 77,246.1 | 66.9 | 24.1 | 368.9 | 46.5 |

Notes.

$[O_3]_{mean}$, average daily (08:−17:00) concentrations of $O_3$ (ppb); $[O_3]_{max}$, average maximum daily concentrations of $O_3$ (ppb); AOT40, cumulative the sum of the differences between the hourly mean ozone concentration in ppb and 40 ppb for each hour of gas exposure; AOT40$^{(150)}$, indicates the accumulated values of AOT40 during the 150-day decomposition experiment (ppb · h); $RH_{mean}$, average daily air relative humidity (%); $T_{mean}$, average daily air temperature (°C); AA, ambient air; EO, elevated $O_3$; $[CO_2]_{mean}$, average air $CO_2$ concentration in OTC ($\mu$mol mol$^{-1}$); DPPFD, average daily photosynthetic photo flux density at the canopy level (mol m$^{-2}$ day$^{-1}$).

nutrient content (mg g$^{-1}$), and nutrient content (mg g$^{-1}$) of leaf litter at the designated time of decomposition, respectively.

Data analyses were performed using SPSS 18.0 (SPSS Inc., Chicago, IL, USA). General linear model (GLM) was used to evaluate and analyse the dynamics of leaf litter mass remaining, changes in C, N, P, lignin, total phenols, condensed tannin and soluble sugar contents with considering the treatments (two levels), decomposition time (continuous variable), as well as their interaction, as independent factors. All data were presented as mean $\pm$ standard deviation. Significant differences between control (ambient air, AA) and elevated $O_3$ concentration (EO) were tested by $T$-test.

# RESULTS

### Changes in chemical compositions in leaf litters of *G. biloba* after exposure to EO

In comparison with AA, EO decreased C content (by 26.1%), but increased slightly N content (by 11.3%) in leaves of *G. biloba* by the end of growing season (Fig. 2). EO decreased C/N ratio (by 10.3%). P content decreased under $O_3$ fumigation. However, changes in C, N, P and C/N were not significant. EO significantly increased the K content by 184.2% ($6.31 \pm 0.29$ vs $17.93 \pm 0.40$, $P < 0.01$) compared with AA (Fig. 2).

In addition, EO decreased the contents of lignin (by 8.6%) and condensed tannins (by 17.5%). By contrast, the difference between EO and AA was not significant (Fig. 2). EO

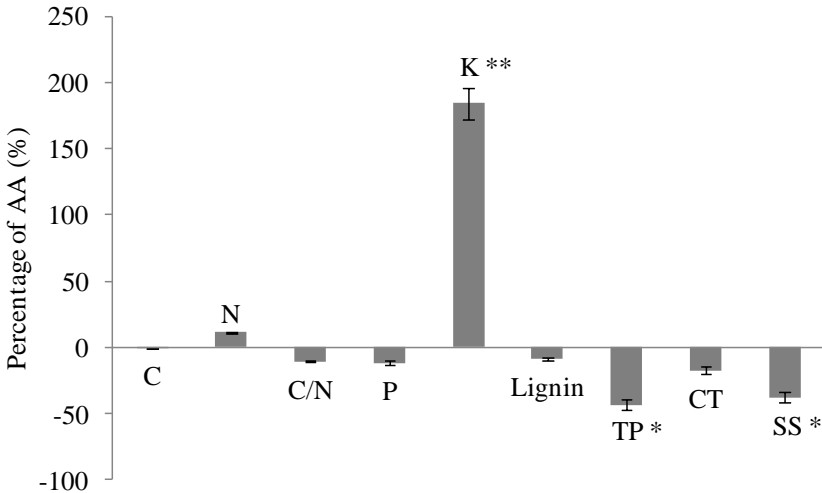

**Figure 2** **Leaf chemistry changes in *G. biloba* as affected by O₃ concentration.**

significantly decreased the contents of total phenolics ($2.82 \pm 0.93$ vs $1.60 \pm 0.44$, $P < 0.05$) and soluble sugar ($86.51 \pm 19.57$ vs $53.76 \pm 2.40$, $P < 0.05$) by 43.3% and 37.9% compared with AA, respectively.

## Dynamics of leaf litter decomposition of *G. biloba* exposed to EO

Based on the Olson's model, the decay constant and the half-life decomposition time of the leaf litters of *G. biloba* under EO were lower than those AA, showing no significant difference (Table 3). Compared with AA, the mass (dry weight) remaining of leaf litters under EO maintained a higher level at early stage of decomposition (before 60-day sampling point), but decreased with decomposition time after 90 days, and it was 5.8% lower than that of AA at 150-day sampling point (Fig. 3A). No significant difference between AA and EO was observed in mass remaining of *G. biloba* leaf litter over time. O₃ fumigation reduced the remaining of C in leaf litter and a significant difference between AA and EO was found after 120 days (EO decreased by 8.3% compared with that of AA) (Fig. 3B). By contrast, N remaining showed a lower level after 90 days under EO than that of AA. However, no significant difference was found in N remaining between treatments during decomposition (Fig. 3C). P remaining in leaf litter showed higher level under EO than AA, and decreased dramatically after 60 days and reached a lowest value at 90-day sampling point (11.2%) (Fig. 3D).

The remaining of lignin in leaf litters of *G. biloba* showed a significant decreasing trend regardless of O₃ treatment during decomposition (Fig. 3E, Table 4, $P < 0.001$). Compared with the AA, the remaining of lignin under EO was quite lower after 120 days, indicating an increasing of lignin decomposition rate. However, no significant effect was observed in the remaining of lignin under EO (Table 4, $P = 0.249$). After 60 days, the remaining of total phenol content showed a significant decrease under EO at each sampling point (Fig. 3F), compared with AA ($P < 0.05$). A significant interactive effect of O₃ concentration and decomposition time was found for the total phenol remaining (Table 4, $P = 0.002$).

**Table 3** Parameters of decomposition rate of *G. biloba* litter leaves exposed to elevated $O_3$ concentration.

| Treatments | Olson exponential decay | $R^2$ | Decay constants (K) | $t_{0.5}$/a |
|---|---|---|---|---|
| AA | $y = 0.8512e^{-1.17x}$ | 0.73 | 1.17 | 0.46 |
| EO | $y = 0.8749e^{-1.39x}$ | 0.84 | 1.39 | 0.41 |

Notes.

AA, ambient air; EO, elevated $O_3$.
No significant difference in each parameter was found between AA and EO.

EO showed no significant effect on the decomposition of condensed tannins during the experiment (Fig. 3G, Table 4, $P = 0.326$). During decomposition, the remaining of soluble sugar showed a significant higher lever at each sampling point except for the 90-day sampling point under EO than AA ($P < 0.05$) (Fig. 3H).

## DISCUSSION

### Effect of elevated $O_3$ concentration on chemical composition of *G. biloba* leaf litter

The results of our study showed that elevated $O_3$ concentration (120 ppb) increased N and K contents in leaf litter of *G. biloba*, but significantly decreased the total phenol content, as well as the C/N and lignin/N ratios. However, the P and lignin contents, as well as the C/P ratio, did not change significantly. Actually, elevated $O_3$ concentration usually changes the chemical composition of plants (*Calatayud et al., 2011*; *Shang et al., 2018*). *Booker et al. (2005)* found that the changes in N and lignin contents of soybean leaf litter increase significantly, while the soluble sugar content significantly decreased under elevated $O_3$ concentration (74 ppb), in agreement with our results in this study. *Parsons, Lindroth & Bockheim (2004)* tested the impact of high $O_3$ concentration (55 ppb) on leaf litter of *Betula papyrifera* for 12 months. The results demonstrated that the contents of N, lignin, and condensed tannins, as well as the C/N ratio of leaf litter, showed no significant change under elevated $O_3$ concentration compared to tests in ambient air conditions.

In our study, the increasing of N content in *G. biloba* leaves after high $O_3$ exposure may be a helpful response to prevent the damage caused by $O_3$ to some extent (*Cao et al., 2016*). High N concentration in leaves could make plants adapt to $O_3$ stress through leaf turnover (*Tjoelker & Luxmoore, 1991*; *Shang et al., 2018*). In addition, the C/N ratio was an important indicator for the degree of coordination of C and N metabolism. Reduction of C/N in plants under elevated $O_3$ concentration showed that the growth of plants was inhibited (*Zheng et al., 2011*), in agreement with our result that C/N ratio decreased under $O_3$ fumigation. *Zhang et al. (2011)* demonstrated that elevated $O_3$ concentration promoted K absorption during growth of plants. Furthermore, K increased N absorption in order to transfer it into proteins under adverse environment. The increase of K contents in our study was possibly one of the reasons why N content increased under elevated $O_3$ concentration.

Plant secondary substances play a pivotal role in scavenging the high level of oxygen species caused by ozone at the end of $O_3$ fumigation (*He et al., 2009*). Generally, the secondary metabolic enzymes such as phenylalanine ammonia lyase (PAL), peroxidase (POD), and polyphenol oxidase (PPO) are involved in the synthesis of secondary

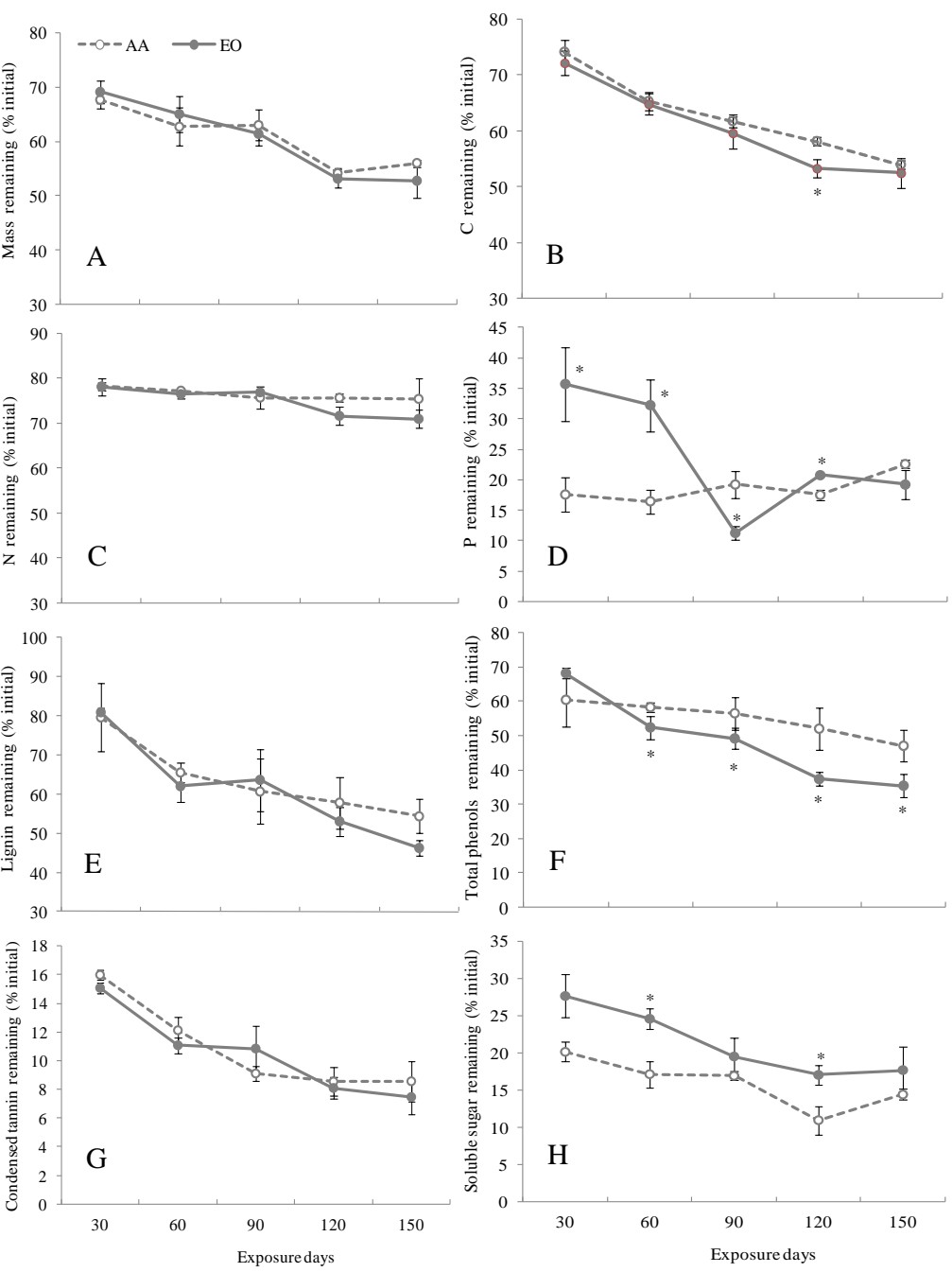

**Figure 3** The remaining of mass (A), contents of C (B), N (C) and P (D), lignin (E), total phenolics (F), condensed tannins (G) and soluble sugars (H) in leaf litter of *G. biloba* under elevated O₃ concentration (EO, 120 ppb) and ambient air (AA, 40 ppb) for 150 days.

**Table 4  Summary of the GLM testing (type III Sum of Squares, in bold values significant at $p < 0.05$) for effects of $O_3$ treatments and decomposition time ($T$) on mass and nutrients remaining of leaf litter in *G. biloba*.**

| | d.f | SS | MS | F | p | | d.f | SS | MS | F | p |
|---|---|---|---|---|---|---|---|---|---|---|---|
| Mass | | | | | | Lignin | | | | | |
| $O_3$ | 1 | 1.31 | 1.31 | 0.23 | 0.637 | $O_3$ | 1 | 43.90 | 43.90 | 1.41 | 0.249 |
| Time | 4 | 963.83 | 240.95 | 42.44 | **<0.001** | Time | 4 | 3,102.30 | 775.57 | 24.87 | **<0.001** |
| $O_3 \times T$ | 4 | 31.34 | 7.83 | 1.38 | 0.276 | $O_3 \times T$ | 4 | 123.04 | 30.76 | 0.99 | 0.437 |
| Residuals | 20 | 113.55 | 5.68 | | | Residuals | 20 | 623.67 | 31.18 | | |
| C | | | | | | TP | | | | | |
| $O_3$ | 1 | 35.78 | 35.78 | 10.04 | **0.005** | $O_3$ | 1 | 302.88 | 302.88 | 16.33 | **0.001** |
| Time | 4 | 1,504.32 | 376.08 | 105.51 | **<0.001** | Time | 4 | 1,995.99 | 499.00 | 26.91 | **<0.001** |
| $O_3 \times T$ | 4 | 15.55 | 3.89 | 1.09 | 0.388 | $O_3 \times T$ | 4 | 446.69 | 111.67 | 6.02 | **0.002** |
| Residuals | 20 | 71.29 | 3.66 | | | Residuals | 20 | 370.91 | 18.55 | | |
| N | | | | | | CT | | | | | |
| $O_3$ | 1 | 20.24 | 20.24 | 4.82 | **0.040** | $O_3$ | 1 | 0.94 | 0.94 | 1.01 | 0.326 |
| Time | 4 | 107.98 | 27.00 | 6.43 | **0.020** | Time | 4 | 225.17 | 56.29 | 60.82 | **<0.001** |
| $O_3 \times T$ | 4 | 36.94 | 9.24 | 2.20 | 0.106 | $O_3 \times T$ | 4 | 8.60 | 2.15 | 2.32 | 0.092 |
| Residuals | 20 | 83.92 | 4.20 | | | Residuals | 20 | 18.51 | 0.93 | | |
| P | | | | | | SS | | | | | |
| $O_3$ | 1 | 200.96 | 200.96 | 25.27 | **<0.001** | $O_3$ | 1 | 217.48 | 217.48 | 58.83 | **<0.001** |
| Time | 4 | 477.90 | 119.48 | 15.02 | **<0.001** | Time | 4 | 366.11 | 917.53 | 24.77 | **<0.001** |
| $O_3 \times T$ | 4 | 796.09 | 199.02 | 25.03 | **<0.001** | $O_3 \times T$ | 4 | 32.24 | 8.06 | 2.18 | 0.108 |
| Residuals | 20 | 159.04 | 7.95 | | | Residuals | 20 | 73.90 | 3.70 | | |

**Notes.**

TP, total phenols; CT, condensed tannin; SS, soluble sugar.

substances (*Tiwari, Sangwan & Sangwan, 2016*). *Luo & Zhang (2010)* found that elevated $O_3$ concentration significantly decreased the lignin contents of plants by inhibiting the activities of PAL, POD and PPO. In our study, the decrease of lignin content in *G. biloba* leaves could be due to the inhibition of activities of relevant enzyme for synthesising lignin by elevated $O_3$ concentration.

Phenolic compounds are the main products of secondary metabolism and are important defence substances in plants. Some research demonstrated that elevated $O_3$ concentrations increased the contents of phenolic compounds in leaves (*Yamaji et al., 2003*; *Peltonen, Vapaavuori & Julkunen-Titto, 2005*). In our previous studies, elevated $O_3$ concentration (80 ppb) significantly increased the total phenolics and condensed tannin contents in leaves of *Quercus mongolica*, leading to the increasing of the antioxidant capacity of plants to $O_3$ stress (*Zhang et al., 2009*). In this study, elevated $O_3$ concentration decreased the contents of condensed tannins and total phenols of *G. biloba* leaves. It could be due to the produce of a large number of free radicals under $O_3$ stress, leading to many antioxidant substances including phenolic compounds were consumed by the end of the growing season (*He et al., 2009*). Therefore, the decreases of condensed tannins and total phenolic content in *G. biloba* leaves may increase the sensitivity of the leaf injuries to $O_3$ (*Yamaji et al., 2003*). Actually, no effect of elevated $O_3$ concentrations on the contents of condensed tannins and phenolic compounds was reported by some studies (*Lavola, Julkunen-Tiitto & Paakkonen,*

*1994*; *Lindroth et al., 2001*), not in agreement with our current results due to the differences in tree species and $O_3$ concentrations to some extent.

Soluble sugar plays an important role in plant metabolism under adverse environments (*Liu et al., 2004*). *Wang et al. (2011)* reported that $O_3$ fumigation (60 ppb, 50% above ambient air) significantly decreased the soluble sugar content of rice in each growth stage. However, *Lu et al. (2012)* found that the content of soluble sugar in the leaves of *Mangifera indica* increased under $O_3$ concentration (50 ppb), but decreased significantly under a high-$O_3$ concentration (200 ppb), in agreement with our result that the soluble sugar contents in leaves of *G. biloba* decreased significantly under elevated $O_3$ concentration (120 ppb). The reason for this was probably due to the accumulation of glycolytic enzymes in leaves, which accelerated the degradation of sugar components (*Tiwari, Sangwan & Sangwan, 2016*).

### Effect of elevated $O_3$ concentration on the decomposition rate of *G. biloba* leaf litter

Elevated $O_3$ concentration not only changes the chemical composition of leaf litter, but also indirectly affects the litter decomposition (*Baldantoni, Fagnano & Alfani, 2011*). In this study, the remaining mass of dry weight of *G. biloba* leaf litter showed significant positive correlation with the C, N, P, and lignin contents as well as the ratios of C/N and lignin/N, regardless of $O_3$ treatment (Table 5). This indicated that the higher the contents of C, N, P, and lignin, the higher the mass remaining, and the lower the decomposition rate (*Bonanomi et al., 2013*). In this study, the mass remaining are larger under elevated $O_3$ concentration than under ambient air at early decomposition stage, which implied that the decomposition of leaf litter slowed down under $O_3$ fumigation. This is consistent with most of the previous studies showing that elevated $O_3$ has adverse effects on litter decomposition (*Parsons, Bockheim & Lindroth, 2008*; *Liu et al., 2009*; *Baldantoni, Fagnano & Alfani, 2011*). At the late stage of decomposition (after 90 days), the mass remaining of leaf litter showed higher level under ambient air than elevated $O_3$ treatment, which indicated that $O_3$ slightly increased the decay rate of leaf litter by impacting litter quality in this study. Indeed, Litter quality is the most important factor affecting mass loss and decay rates of nutrients (*Bonanomi et al., 2010*).

In fact, $O_3$ exposure has not always led to reduction in litter decomposition rate of tree species (*Scherzer, Rebbeck & Boerner, 1998*; *Kainulainen, Holopainen & Holopainen, 2003*). In our recent study, we observed that N mineralization and lignin degradation in leaf litters of *Q. mongolica* under elevated $O_3$ concentration (120 ppb) were inhibited during early stage of decomposition, but promoted at later stage of decomposition (*Su et al., 2016*). Indeed, as the major component in leaf litter, lignin has a complex structure and is difficult to decompose (*Peng & Liu, 2002*). During the decomposition in this experiment, the remaining rates of lignin, condensed tannin and total phenols were lower under elevated $O_3$ concentration than under ambient air at late decomposition stage, which implied that elevated $O_3$ concentration promoted the decomposition of leaf litter. The slight promotion of the decomposition rate of leaf litter mainly resulted from the decrease of leaf litter quality induced by $O_3$ fumigation. Besides, exposure of $O_3$ at the late decomposition stage of the

**Table 5** Spearman correlations coefficients of remaining mass to initial with nutrient contents dynamics during decomposition of *G. biloba* leaf litter.

|  | Treatments | Remaining mass | C | N | C/N | P | Lignin |
|---|---|---|---|---|---|---|---|
| Remaining mass | −0.07 |  |  |  |  |  |  |
| C | −0.07 | 0.96** |  |  |  |  |  |
| N | 0.12 | 0.33* | 0.28 |  |  |  |  |
| C/N | −0.17 | 0.91** | 0.96** | 0.17 |  |  |  |
| P | 0.28 | 0.79** | 0.79** | 0.48** | 0.66** |  |  |
| Lignin | −0.36* | 0.66** | 0.66** | 0.28 | 0.75** | 0.36* |  |
| Lignin/N | −0.32 | 0.63** | 0.67** | 0.19 | 0.78** | 0.34* | 0.95** |

**Notes.**
*$P < 0.05$.
**$P < 0.01$.

experiment might accelerate the oxidation and decomposition of secondary metabolic substances (*Su et al., 2016*).

Unlike the remaining of lignin, condensed tannin and total phenols, the remaining of soluble sugar showed higher values under elevated $O_3$ concentration than ambient air at any sampling point during litter decomposition. The lower decay rate observed for ozone-exposed leaves could be due to changes in both structural and functional molecules that we did not study here or decreases in the activities of enzymes relating to carbohydrate metabolism under $O_3$ exposure (*Peace, Lea & Darrall, 1995*). In addition, the higher value of carbohydrate remaining was maintained under $O_3$ fumigation during decomposition and becoming much smaller difference between AA and EO by the end of the experiment, which might result from inhibiting effect of $O_3$ on decay rates of leaf litter with high initial values at early decomposition stage. The similar results were observed in some previous studies (*Fioretto et al., 2005*; *Liu et al., 2009*).

## CONCLUSIONS

In this study, we found that elevated $O_3$ showed no significant impact on chemical compositions and decay rates of *G. biloba*, although it decreased the contents of C, P, C/N ratio, lignin, total phenols, condensed tannins, and soluble sugars in *G. biloba* leaves by the end of gas fumigation. In fact, $O_3$ fumigation slightly inhibited the decomposition of *G. biloba* leaf litter at the early stages of decomposition, but increased decomposition rate at late stages of this experiment. During the whole decomposition, the losses of the nutrients in leaf litters of *G. biloba* showed significant seasonal differences regardless of $O_3$ treatment. Rising atmospheric $O_3$ concentration will likely elicit species-specific effects on litter production and decomposition in urban forests, retarding decay rates in some species such as birch (*Parsons, Bockheim & Lindroth, 2008*), while potentially exerting little effect on others. Furthermore, elevated $O_3$ will not exert its influence on litter decay rates in isolation from other factors including soil microorganism, although the results from our study showed that $O_3$ in current concentration in this study had little (not significant) effect on decomposition of *G. biloba* leaf litter. Our results indicated that the ground-level $O_3$ concentrations in some cities of China could significantly alter the chemical compositions

and decomposition rates of at least one deciduous gymnosperm tree species once suffering from long-time exposure of higher $O_3$ concentration. These changes are likely to have important implications for our understanding of the processes regulating the storage and emission of C from urban forest ecosystems under climate change. Therefore, further research for the long-term $O_3$ exposure of urban trees (only one growing season in this study) is necessary to determine the nutrient cycling and sustainability of main tree species with different ages including *G. biloba* in urban forest, where $O_3$ is one of the most widespread of all the gaseous pollutants in urban area.

### Funding

This work was jointly supported by the National Natural Science Foundation of China (NSFC, 41675153, 31270518 and 31170573), the Key development program of the Chinese Academy Science (KFZD-SW-302-01) and the Key Project of NSFC (90411019). The funders had no role in study design, data collection and analysis, decision to publish, or preparation of the manuscript.

### Grant Disclosures

The following grant information was disclosed by the authors:
National Natural Science Foundation of China: 41675153, 31270518, 31170573.
Chinese Academy Science: KFZD-SW-302-01.
Key Project of NSFC: 90411019.

### Competing Interests

The authors declare there are no competing interests.

### Author Contributions

- Wei Fu performed the experiments, analyzed the data, contributed reagents/materials/-analysis tools, prepared figures and/or tables, authored or reviewed drafts of the paper, approved the final draft.
- Xingyuan He and Wei Chen conceived and designed the experiments, authored or reviewed drafts of the paper, approved the final draft.
- Sheng Xu analyzed the data, prepared figures and/or tables, authored or reviewed drafts of the paper, approved the final draft.
- Yan Li analyzed the data, prepared figures and/or tables, authored or reviewed drafts of the paper, approved the final draft, search for the references.
- Bo Li conceived and designed the experiments, performed the experiments, contributed reagents/materials/analysis tools, authored or reviewed drafts of the paper, approved the final draft, revised the paper.
- Lili Su performed the experiments, contributed reagents/materials/analysis tools, prepared figures and/or tables, authored or reviewed drafts of the paper, approved the final draft.

- Qin Ping performed the experiments, analyzed the data, prepared figures and/or tables, authored or reviewed drafts of the paper, approved the final draft.

## Data Availability

The raw data is provided in the Supplemental Files.

## Supplemental Information

Supplemental information for this article can be found online at http://dx.doi.org/10.7717/peerj.4453#supplemental-information.

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
