# Peer review of "Changes in nutrients and decay rate of Ginkgo biloba leaf litter exposed to elevated O3 concentration in urban area"

_PeerJ, doi:10.7717/peerj.4453_

## Round 0.1 · original submission · Major Revisions

Reviewers have added detailed comments and suggestions you should take into account when preparing your revised manuscript. In particular, reviewer #1 noted that research question is well defined but more information on its relevancy must be added, whereas reviewer #2 asked for an accurate revision of the statistical analysis, in order to make the results much more robust and comparable.

Reviewer 1 ·

Basic reporting

English must be improved
More information on results of other studies should be added and an hypothesis too
The structure is OK
An hypothesis is missing

Experimental design

It fits with the scope of the journal
Research question well defined but more information on its relevancy must be added
IMethods are well described

Validity of the findings

Data and statistics are correct.
Conclussions should be improved

See General comments to the authors for more comments....

Additional comments

The paper addresses the question of how ozone affects the chemical composition and decomposition of Ginko biloba litter. The topic is in principle interesting as there are not many studies on this issue. The experiment was carried out in Open Top Chambers in China. However, there several relevant questions that makes the paper not acceptable for publication in its current form. Here there are some recommendations:
Some English expressions are somewhat awkward, and I strongly recommend that the paper is revised by a native speaker. Some examples: line 31: has a toxicity to plants that is widely concerned; line 67: every treatment was repeated, maybe: every treatment was replicated; line 69: management measurements…were consistent, maybe: Management measures such as irrigation were the same for the two treatments; line 95: through use, maybe: through the use; line 157: changed insignificantly compared test in ambient air (sentence not clear); line 153: , as is in agreement (better: which is in agreement); lines 174-178: unclear sentences; line 189: thereof is conducive (unclear meaning).
Lines 6-7: Is the topic of this paper forest ecosystems or urban vegetation?. The meaning of “Profoundly influenced” is not clear, what is the type of effect?, or remove.
Line 8: Ginkgo biloba was -> maybe: Ginkgo biloba saplings were used
Line 11: improved->increased
Line 11: reduced->and reduced
Line 16: of->maybe: caused by
Line 28-29: Although this is the case for Asia, it is not true for Europe, as the increase has stabilized recently (see Cooper et al., 2014. "Global distribution and trends of tropospheric ozone: An observation-based review." Elementa: Science of the Anthropocene 2, 000029).
Line 50: A clearer hypothesis should be proposed, rather than “change”, better put increase or decrease. The hypothesis should be proposed after introducing previous studies. Based on previous studies, it should be expected that there is an increase or decrease. Then, test if that hypothesis is correct or not.
Line 65-66: The fumigation regime appears unrealistic for urban trees as 120 ppb is quite a high value sustained for about 4 months (please, add the dates). Please, justify what such high values were used.
Line 76: what is the regular litter bag method?. Please, add a reference
Line 99: Can you add a reference for this model?
Line 118-121: add that the changes in N and P were not significant.
Line 123: obviously->significantly
Line 144: at 0.05 and 0.01 levels->better refer to significance here, not to levels
150: Calatayuda is Calatayud (check also other parts of the paper)
150: In this context the following paper by Shang et al. (2018) can be relevant: "Elevated ozone affects C, N and P ecological stoichiometry and nutrient resorption of two poplar clones." Environmental Pollution 234, 136-144.
Line 202: please, add supporting references
Discussion: Several issue have not been well explained
O3 usually reduces N content of older leaves as there is a translocation of this nutrient to upper leaves which increases senescence. Why N increases in the litter?. Is it the case that more litter is produced?. Please, explain this.

Other important improvements are:

The discussion is descriptive of the results and there are comparisons with other studies but one does not get a complete picture of the processes behind such results. Please, try to give a more comprehensive picture of the processes.
Conclusion: Besides summarizing the results, what is the relevancy of these results for urban forests?. A wider scope is needed in the discussion.

Reviewer 2 ·

Basic reporting

Language: although generally clear, the manuscript language could be improved at several points towards professional standard for scientific writing. Specific comments to the sections requiring language revision are reported below.
Literature references: insufficient field background is provided, particularly at specific points in the Introduction and Discussion sections, where substantially relevant information about previous evidence is missing. Please see specific comments below.
Article structure is mostly satisfactory. Specific comments aimed at improving figure and table style and quality of presentation are reported below for each item. Raw data are also presented as supplemental excel files, which is appreciable, but not described by metadata.

Experimental design

Some major flaws in the study rationale and experimental design are highlighted in the comments below (see sections 3 and 4).
Research question are potentially relevant & meaningful, although not well defined. It is not explicitly stated how research could fill the knowledge gap about O3 effects in urban forests.
Methods are not described with sufficient detail & information to replicate.
More details in comments below (section 4)

Validity of the findings

Statistical analyses should be re-done.
The absence of a true control weakens the authors’ conclusions.
Raw data are provided as excel files, which include all graphs presented in the manuscript figures. However, these are based on aggregated data (means and standard deviations) that are not linked to raw data by formulas, so that it is not possible to assess the correspondence between raw data and data presented in the manuscript figures. Possibly easily addressable by providing appropriate metadata (which raw data are used for each data point in each figure?)
Conclusions are concise, but at the current state of the manuscript they are not completely supported and in the case of litter decay acceleration by high O3 concentration could even be mistaken.
Speculation is widely present in the Disucssion section, not always identified as such.
More details in comments below (section 4).

Additional comments

Title
The manuscript title is not informative of the study results. As the study is about leaf litter, this should be included in the title. Suggestion: “Exposure to elevated O3 concentration enhances nutrient content and decay rate of Ginkgo biloba leaf litter”

Abstract
The abstract section could be largely improved, by including a more effective reference to the background, clearly stating the tested hypothesis, and adding missing information. Please revise this section along the line of the following specific comments:
LL 6-8: “has profoundly influenced” does not clarify the effects of exposure to O3 on forest ecosystems. Please rephrase including a general description of these effects. Which forest ecosystem are more affected, which response metrics are affected, by which magnitude and direction?
L 8: “still less known” perhaps better “still poorly known”
LL 9-10: “to investigate the impact…” at this point the study objective is not clear. The authors should consider to clearly define the scientific hypothesis underlying their study. What are the expected impacts on litter chemical composition and decay rate? See comment to Introduction LL 49-53.
Language issue: please consider adding “concentration” to “elevated O3” throughout the text.
The authors here refer to “leaf litters of the urban tree”, inconsistent to the preceding background sentences that refer to “forest litters”. In facts, the manuscript reports on a manipulative experiment in an arboretum. This should be clearly stated.
Information on experimental conditions and the methods used to assess litter chemical composition should be included. At this point, it is not clear what the authors mean by “OTCs simulations”.
LL 11-12: some quantitative data (e.g. mean and sd of treatments, p-values for statistical significance) on the litter content of N, K, phenols, tannins and sugars should be included. Please consider rewording according to the following suggestion: “Exposure to high O3 concentration, compared to the ambient air control, enhanced the litter content of N (mean ± sd vs mean ± sd, P < xxx) and K (mean ± sd vs mean ± sd, P < xxx), while reducing that of total phenols (mean ± sd vs mean ± sd, P < xxx), condensed tannins (mean ± sd vs mean ± sd, P < xxx) and soluble sugars (mean ± sd vs mean ± sd, P < xxx).”.
Please also consider adding a comment about the result of lignin shown in Fig. 3.
LL12-13: “Furthermore, elevated O3 significantly decreased the residual rate of dry weight by 11.9% of G. biloba litters (P < 0.05)”. Awkward sentence. Please reword, along the line of the following suggestion, more consistent with results presented in Fig. 2: “Furthermore, percent mass remaining in litterbags after 150 days of exposure to high compared to low O3 concentration was 11.9% lower (P < 0.05)”.
Please note: a) the use of “percent mass remaining”, which is the established definition Y = At/A0, in place of “residual rate” or “residual ratio”, that the authors inconsistently use throughout the manuscript; b) the inclusion of experimental conditions under which the result was observed. However, about this specific result, see comment to Fig. 2 below.
L 14: What are the “nutrient components”? Consider using simply “nutrients”. Possible rewording: “Differently, net nutrient release from litter during decomposition was unaffected by O3 treatment.”.
L 15-17: current version: "as is mainly related to"; prudent alternative version "may be related to" (or something like that). I recommend writing prudently, considering that the authors only showed that lignin content decrease with litter decay under high O3 concentration, but they did not directly test the occurrence of lignin oxidation, nor reported previous evidence about this possible explanation for the observed enhanced litter decay. Speculation could be admitted, but should be identified as such. Please also consider discussing results on litter chemical composition as affected by treatments.

Introduction
This section should be re-arranged in order to improve the logical order of presentation of different arguments. Please follow specific comments below.
LL 30-31: This sentence is scarcely informative. What is the point in reporting that O3 toxic effects on plants are widely concerned, when such effects are not reported? The authors should either insert a synthetic sentence that describes O3 toxicity, or delete the sentence.
LL 31-32: Again, “direct or indirect influences” is meaningless without specifying which influences we are speaking of, and which are direct or indirect. The authors should either insert a synthetic sentence that describes the “influences”, or delete the sentence. In case the sentence is not removed, please consider replacing “material cycles” with “biogeochemical cycles”. “Catalayuda” is misspelled. Please use the correct author surname “Catalayud”.
LL 34-44: this paragraph is a mix of different topics whose order of presentation is hard to follow, and lacks of appropriate reference to the literature. The paragraph should be re-written leaving out topics out of context, such as climate change, increased CO2, nitrogen depositions. I suggest to include the following argumentations, with appropriate citations where needed:
Effects of O3 on forest ecosystems productivity and feedbacks have been widely investigated worldwide (reviews in Chappelka & Samuelson, 1998; Paoletti, 2006; de Bauer and Hernández-Tejeda, 2007) and recently synthesized (Wang et al. 2016). Differently, O3 effects on litter decomposition are much less known. Indirect evidence comes from litter photodegradation studies in semi-arid and arid ecosystems (e.g. Austin & Vivanco, 2006 and following). Among the few studies carried out in natural forest ecosystems, Parsons et al. (2008) showed that, over a 23-months observation period on leaf litterbags of aspen and birch reciprocally transplanted to separate substrate quality from environment effects, increasing O3 concentration by fumigation slowed down both aspen and birch litter decay rate, exhacerbating the effects of elevated CO2 concentration, but accelerated birch litter decay under ambient CO2. A negative effect of O3 fumigation on litter decay rate was also observed for holm oak leaf litter in Mediterranean forest (Baldantoni et al. 2011). Such observations were explained by CO2- and O3-mediated changes in litter chemistry, particularly carbohydrates, nitrogen, and tannins. O3 effects on litter decomposition in urban forests have not yet been explored. Filling such gap is particularly important, as in urban ecosystems, where tropospheric O3 concentration can be very high due to photochemical air pollution, urban trees play a fundamental mitigating role (e.g. Manes et al. 2012). Their leaf litter, if decaying faster when exposed to high O3 concentration, would improve soil chemical properties and promote nutrient cycles, therefore affecting the sustainable development of urban areas (Nikula et al., 2010; Xu et al., 2012).
Cited references:
Austin, A.T., Vivanco, L., 2006. Plant litter decomposition in a semi-arid ecosystem controlled by photodegradation. Nature 442, 555-558.
Calfapietra, C., Fares, S., Manes, F., Morani, A., Sgrigna, G. and Loreto, F., 2013. Role of Biogenic Volatile Organic Compounds (BVOC) emitted by urban trees on ozone concentration in cities: A review. Environmental pollution, 183, pp.71-80.
Chappelka, A.H. and Samuelson, L.J., 1998. Ambient ozone effects on forest trees of the eastern United States: a review. The New Phytologist, 139(1), pp.91-108.
de Bauer, M.D.L. and Hernández-Tejeda, T., 2007. A review of ozone-induced effects on the forests of central Mexico. Environmental Pollution, 147(3), pp.446-453.
Manes, F., Incerti, G., Salvatori, E., Vitale, M., Ricotta, C. and Costanza, R., 2012. Urban ecosystem services: tree diversity and stability of tropospheric ozone removal. Ecological applications, 22(1), pp.349-360.
Paoletti, E., 2006. Impact of ozone on Mediterranean forests: a review. Environmental Pollution, 144(2), pp.463-474.
Parsons, W.F., Bockheim, J.G. and Lindroth, R.L., 2008. Independent, interactive, and species-specific responses of leaf litter decomposition to elevated CO2 and O3 in a northern hardwood forest. Ecosystems, 11(4), pp.505-519.
Wang, B., Shugart, H.H., Shuman, J.K. and Lerdau, M.T., 2016. Forests and ozone: productivity, carbon storage, and feedbacks. Scientific reports, 6, p.22133.

LL 46-48: please rephrase avoiding the use of passive voice. E.g.: In recent years, we assessed the effects of elevated O3 concentration on the leaf eco-physiology of this species (He et al., 2009; Lu et al., 2009; Li et al., 2011). Here, we aim to complement such previous studies with a manipulative experiment testing O3 effects on G. biloba leaf decomposition and chemical features.

LL 49-50: expected outcomes are too broadly reported. The authors should specify at least the expected effect direction, or even effect magnitude, on the base of previous evidence, their own argumentations, or both. In this respect, it has to be considered that the only previous experiment directly testing O3 fumigation effects on leaf litter decomposition (Parsons et al. 2008, see above), though carried out on different species and under different conditions compared to the present study, produced different results for different conditions. This should be explicitly reported, and expectations clearly and unambiguously defined and justified.

LL 51-53: Main objective 1 is well defined, although language can be improved, e.g. replace “check the changes in” with “assess changes of ”. Main objective 2 should be more accurately defined, such as: 2) to assess leaf litter decomposition rate of this urban tree, as affected by O3 fumigation treatment.


Materials & Methods
L 57: “where there is” maybe nearby?
L 58: you mean a mean elevation? Is it the region that covers the urban forest, or vice-versa?
LL 59-63: please add appropriate reference for climate classification and data. Consider replacing “per annum” with “yearly”.
L 65: Replace “A total of two” with “Two”.
LL 66-67: A description of the fumigation system, or at least a reference to one of your previous studies where such system was applied, would be appreciated. Treatment naming is confusing. If you applied O3 fumigation at 40 ppb, I do not see the point of naming such treatment “ambient air”, unless you show that ambient air in natural conditions contains a constant ozone concentration of 40 ppb. Consider using LOC (Low Ozone Concentration) and HOC (High Ozone Concentration) to name the treatments. Your experimental setup suffers the lack of an untreated control, which prevents from assessing litter chemistry composition in truly ambient air, that is in not fumigated leaf samples. Indeed, based on your experimental setup, deviations from natural conditions attributable to ozone treatments cannot be assessed, which weakens your conclusions.
LL 67-75: Treatment repetition means that you replicated the treatment over three different time periods. This is not the case, isn’t it? If I correctly understand your experimental design, you just considered three independent replicates of both treatments. Please confirm and change the text accordingly. Please add information on seedling history before planting: five years are missing from the story, and could affect your results. “management measures were consistent”, poor language and incomplete information. Do you mean that cultivation practices were either consistent among treatments and replicates or consistent over time? Or both? Please add quantitative information on irrigation rate and timing and other cultivation practices, if any. I would not use “seedlings” for five years old plants and “trees” for five years and 60 days old plants. Please use consistently either of the two. Replace “fumigated by” with “fumigated with”. “All yellow leaves…” you mean senescent standing leaves or fallen leaves? Please clarify. “These yellow leaves…” you mean that you physically divided (cut?) each leaf into two parts? Please reword to clarify, e.g. “leaves from each OTC were subdivided into two pools”. Replace “initial chemical composition” with “chemical composition of undecomposed litter”. Remove “sufficiently” and replace with information on drying time. Replace “next year” with “the following year”. Where and in which conditions was the litter material stored before the decomposition experiment?
LL 76-88: Remove “regular” (is there any “irregular” litter bag method?) and add appropriate reference for this method and possible modifications. “each of the 15 bags was placed into each OTC”: presumably poorly written, as literally meaning that litterbags were moved from one OTC to another, which is unrealistic. How many OTC were used for the decomposition experiment? And how many bags per OTC and in total? What do you mean by “close to the soil surfaces”? Please indicate if bags were placed above or below the soil surface and at which height/depth from the ground level. “in a random order” is also poorly written. A correct random and reproducible procedure should be described as “N pools of K litterbags were randomly selected out of a total number of N * K, and each pool was randomly assigned to one of N different OTCs, where litterbags were randomly placed X cm above/below the soil surface.” Considering the soil depth (i.e. 0-10 cm), I would rather speak of “topsoil”. Replace “deep” (adjective) with “depth” (noun). Where do data in Table 1 comes from? Either you collect OTC topsoil samples and measured chemical-physical properties of collected samples, and in this case you should add information on topsoil sampling and chemical-physical measurements, or those data were previously published, which is not an issue, but in this case it must be appropriately referenced to a paper where methodological information can be found. The same applies to micro-environmental data reported in Table 2. “These litter samples” avoid the use of “these”. About pulverization and filtering, please add information on mesh size as related to sample particle size.
LL 89-97: I would replace “Measurements”, which is too generic, with “Chemical analyses” in the subsection title. Which elemental analyzer was used? Please indicate model name and producer, so that information on precision and accuracy can be retrieved. Please specify with more detail or appropriately reference the methods used to determine lignin, phenols, condensed tannins and soluble sugar content. In the case of lignin, please note that the accuracy of UV/Vis spectra can be highly affected by the type and structure of lignin (i.e. relative abundance and linking of H, S and G residues), the used solvent, and the pH of the solution. Please justify your methodological choices with respect to alternative methods (i.e. for lignin they range from spectroscopic techniques, such as ICP-Mass spectrometry or reflectance IR spectroscopy, to simple and fast analytical methods, such as proximate analysis or acetyl-bromide method.
LL 98-104: “Decomposition rate of litters was represented by residual rate (%) of litter and were calculated by using the Olson litter decomposition index model.” Consider rephrasing as “Litter mass remaining over time, expressed as percentage of the initial value, was calculated according to the well-known simple exponential model formulated by Olson (1963).” (see comment above to the Abstract section LL 12-13). Don’t need to define the Euler’s number. Please note that if you express Y as monthly, then units of measure of t and k are months and months-1, respectively, not days. Also note that the “fitting parameter” a is not included in the Olson’s model. By the way, according to your equation, at t=0, Y = ae-kt = 1, with e-kt =1 and a=1. Therefore, a cannot be a parameter to fit. Half-life time equation is also wrong, as well as the subscript notation: the correct conventional notation is t½ or t0.5 and it is calculated as t½ =ln(2)/k (note that k is a positive number, as –kt is negative and t is positive). Complete decomposition time is questionable, and in the case of this manuscript is useless, given the short experiment duration.
Cited reference:
Olson, J.S., 1963. Energy storage and the balance of producers and decomposers in ecological systems. Ecology, 44(2), pp.322-331.
LL 105-112: The authors cite a paper by Brandt et al. (2010) to justify the mass balance equation used to calculate “release of nutrient element assay” based on “residual rate of elements”. However, neither such equation, nor the terminology is reported in the cited paper, where “% of initial N” is calculated (Brandt et al. 2010, page 771, Fig. 5). Please justify.
LL 113-115: description of performed statistics is inaccurate, likely as being too short. Based on what is reported, the authors tested differences between two means (AA vs EO) with a one-way ANOVA plus LSD (post-hoc) test, which is basically wrong, as ANOVA is intended to test differences among three or more treatment levels. Also, given the small sample size (presumably N=3, which is the number of OTCs, for each of the two treatments), ANOVA assumptions of normality and homoscedasticity cannot be tested, therefore leaving room for using a non-parametric test for two independent samples (e.g. Mann-Whitney-Wilcoxon test). However, given that the dependent variables were measured for both treatments at 6 sampling dates (i.e. 30, 60, 90, 120 and 150 days of decomposition, plus initial values in undecomposed litter samples), using a one GLM model for each dependent variable (litter mass remaining, N, P, etc.) would be more appropriate, including fumigation treatment as a fixed effect with two levels, and litter age as a continuous covariate. See e.g. Bonanomi et al. 2013, Soil Biology & Biochemistry 56: 40-48, 2013 as an example of this type of statistical analysis applied to litterbag decomposition under different treatments.

Results
LL 118-125: The authors should re-analyze their data based on the comments reported above, and re-write the Result section accordingly.
LL 126-138: Please avoid wordy expression such as “the Olson attenuation index model”. Simply use “Olson’s model”. Remove “complete decomposition time”. The use of such indicator is a bit stretched: no litter decomposes completely, especially in a time frame as that tested in this manuscript. What the authors mean by “small differences (Table 3)”? Are those differences significant or not (easy to test by an ANCOVA model)?
About the rest of the result section, please see specific comments on Tables and Figures as follows:
Table 1. Besides comments on methods for assessing topsoil features described above, I would suggest to include information on additional features possibly affecting litter decomposition (e.g. texture, salinity, organic matter content, etc.). Moreover, as a table footnote specifies that soil data are mean of 3 replicates, standard deviations should be added. Given the very similar feature values, differences in topsoil properties between the two treatments are likely not statistically significant. This should be explicitly stated, at least in a footnote, since apparently it was not directly tested.
Table 2. Not sure that all this information is properly presented in its current form. I would suggest to move these data in one Figure with two panels, one for each treatment, showing time profiles throughout the observation period for O3 (both concentration and AOT40) and microclimatic variables (T, RH). Such figure would be more informative of time-dependent variability of the experimental conditions, currently not clear from the Table. Among microclimatic variables, I would add light irradiance time profile (possibly as a further or supplementary figure). Indeed, irradiance effects interacting with O3 treatments might not be excluded in relation to possible litter photodegradation. These effects may also vary in relation to canopy cover variability at the litterbag placement sites, therefore I would be pleased to see irradiance time profiles (or at least canopy cover data) for each litterbag of each OTC.
Figure 1. Consider rewording the caption as “Leaf chemistry changes in Ginkgo biloba as affected by O3 concentration. Data refer to mans and standard deviation of the content of carbon (C), nutrients (N, P, K), C-to-N ratio, total phenolics (TP), condensed tannins (CT) and soluble sugars in leaf samples (N=3) of the target specie after 3 months of daily exposure to high O3 concentration (120 ppb), expressed as percentage of corresponding data observed at lower O3 concentration (40 ppb). Significant changes are shown (*, P < 0.05; **, P < 0.01).”
Figure 2. I am puzzled by the fact that the authors base most of their conclusion about O3 accelerating effects on litter decay on this figure. Indeed, the authors correctly report higher mass remaining in the 40 ppb treatment compared to 120 ppb after 150 days of decomposition (but see above about the need to re-analyze the data to properly assess statistically significant effects). However, it should be equally stressed that litter mass loss dynamics for the two treatments are not different over 120 days, indicating a large independence of decomposition from O3 for most of the observation period. Moreover, it is not correct to describe the difference observed after 150 days as an “acceleration” of litter decay under high ozone concentration. Indeed, in the 30-days period from 90 to 120 days, litter mass remaining in EO passes from 61.4% to 51.5% of the initial value, for a mass loss of 10%, while in the 30-days period from 120 to 150 days, litter mass remaining passes from 51.5% to 49.5%, for a mass loss of 2% (see supplemental raw data file Data1.xls). These data indicate that litter decay in EO slows down in the last 60 days. But what is more surprising, is that at the corresponding dates AA treatment shows a mass increase! In other words, the result does not show an increase in litter decay rate under elevated O3, but that litter material does not decompose, and even increases in mass, under low O3 concentration. Such observation indicates either a possible inclusion of exogenous material in the litterbags (microfauna? microbial biomass? other?) or a misprocessing of the sample. How the authors explain such unexpected result? This is particularly relevant, as the manuscript conclusions mostly rely on this point.
Minor comment: not sure about the asterisk indicating significant difference of litter C content in treatments after 150 days. Since standard deviation bars overlap, and sample size is low (3? 6?), standard errors and confidence interval widths should not be so small to produce a significant difference. Please double check after re-analyzing data according to the comment at LL 113-115 above.
Figure caption should be re-worded as “Mass remaining and C, N and P content in leaf litter of Ginkgo biloba daily exposed to either higher (HO, 120 ppb) or lower (LO, 40 ppb) O3 concentration for 150 days. Data refer to mean and standard deviation of N replicated litterbags for each treatment. Significant treatment-dependent differences are shown (*, P < 0.05; **, P < 0.01).”
Figure 3. As this figure is completely consistent with Figure 2, in terms of layout and content, I would merge the two figures into one, being a 2 x 4 panel. Caption of the new figure should be arranged as commented for Figure 2 above.

Discussion
LL 149-162: Subsection title, “composition” is singular. The sentences at LL 157-160 should be synthesized and moved at the beginning of the paragraph, in order to briefly resume your results prior to a comparison with previously reported evidence.
“Catalayud”, see same comment about the Introduction section. Delete “of chemical composition” at L 151, as well as “as in” and “in this study” at L 153. Use past tense when reporting previous observations (i.e. “demonstrated” at L 155). “…compared test in ambient air.” Change with “…compared to tests in ambient air conditions”. The use of “obviously” when commenting results about N and K should be either justified, or avoided.
The comment at LL 160-162, in its current formulation, is too broad and obvious: of course experiments carried out under different conditions can produce different results. The interesting point here is to what extent (and direction) such conditions and results are different and how the different conditions could explain the observed differences in the results. Any mechanistic hypothesis?
LL 163-165: this is a repetition of LL 157-158. Could the authors extend and clarify their hypothesis about “litter stress resistance” at L 165-166?
LL 166-169: potentially interesting, but requires a more accurate formulation (e.g. what is intended for “leaf transformation mechanisms”?) and more logical argumentation. Btw, in the cited paper by Maurer et al., intraspecific variability in leaf N content, as affected by different fertilization regimens, affect resistance to O3 damage. Are there possibly cause-effect misinterpretations? Maurer’s observation is about a leaf trait (high N content) that (possibly) lessen resistance to O3 damage. Here the authors show that exposure to O3 causes higher leaf N content. Not exactly the same cause-effect relationships. Could the authors clarify?
LL 179-180: “some”, “certain”… either specify which, or delete the sentence.
LL 203-220. This paragraph suffers a substantial lack of reference to previous, well established knowledge on litter decomposition as related to C, N and lignin dynamics. It should include at least the argumentation suggested above when commenting the introduction (see comment at LL 34-44) and discuss the results in comparison to previous evidence (Parsons et al 2008, Baldantoni et al. 2011), which is apparently contrasting with the presented results. I suggest to start with a statement resuming your results (e.g. a reduced version of LL 205-209, but please avoid to explain the interpretation of correlation results). About the relationship between litter N content and decay rate, many precious references are available after Melillo (1982). Please have a look to Berg, B., and McClaugherty, C., 2008. Plant Litter: Decomposition, Humus Formation and Carbon Sequestration, second ed. Springer-Verlag, Berlin, Heidelberg. As this source, being a book, may be possibly unavailable, a good discussion of C and N topics can be also found in the works by Bonanomi et al.: 2010, Plant and Soil, 331(1-2), pp.481-496; 2013 Soil Biology and Biochemistry, 56, pp.40-48, 2014 Plant Soil (2014) 381:307-321.

Conclusion
As a conclusive remark, the authors could comment on the relevance of their findings within the context of decomposition studies, including possible effects of increasing tropospheric ozone concentration on the C budget at ecosystem level, at least in the urban forest where the study site is located. In other words, would it be possible to tentatively estimate or at least discuss the increase in CO2 emissions due to litter decomposition as affected by higher O3 levels? Although decisions are not made based on subjective determination of novelty and interest of the study, adding such information would certainly provide a remarkable added value to the study, making it likely interesting for a broader audience.

---

## Round 0.2 · accepted · Accept

I have now examined your responses to reviewers' remarks and I feel that the revised version of your paper meets their recommendations for more details about statistical analyses and for a more accurate explanation of the relevance of your work.